# Capital market liberalization and corporate environmental performance: Evidence from the Shanghai–Hong Kong Stock Connect

Qi'an Zhong * 

School of Business, Suqian University, Suqian, China

* 21074@squ.edu.cn

## Abstract

The implementation of Shanghai-Hong Kong Stock Connect marks the maturity of China's capital market, and the effect of the implementation has been the focus of academic attention. Based on this quasi-natural experiment, We select 3248 samples of heavily polluting enterprises listed in China in 2010–2020 to examine the impact of capital market liberalization and on corporate environmental performance.The results show that capital market liberalization significantly improves the environmental performance of heavy polluting enterprises. The results of the heterogeneity analysis indicate that the positive effect varies across firms with different ownership and internal controls. Finally, mechanism analysis results find that capital market liberalization promotes the environmental performance of heavily polluting firms by increasing environmental assets,reducing stock price volatility,and improving the quality of information disclosure.

**Editor:** Ömer Tuğsal Doruk, Adana Alparslan Turkes Science and Technology University: Adana Alparslan Turkes Bilim ve Teknoloji Universitesi, TÜRKIYE

**Data Availability Statement:** The data underlying the results presented in the study are available from DataDryad via the following URL: https://

## 1. Introduction

Capital market liberalization has always been a highlight of the financial reform in China. Relevant policy proposals, ranging from QFII and RQFII to connectivity, also indicate that China's financial market is gradually maturing. Among them, connectivity has been adopted as crucial to the development of China's financial market,such as Shanghai-Hong Kong Stock Connect (SH-HKSC), and Shenzhen-Hong Kong Stock Connect (SZ-HKSC). On this basis, The National Development and Reform Commission (NDRC) formulated the "Relevant Measures on Further Expanding the opening of the Financial Industry"to further relax the conditions for foreign investment, shareholding ratios and so on. Given the opening of the capital market, China's A-shares have been included in the MSCI, FTSE, Russell, and Standard Dow Jones indices [1]. Hence, China's capital market has become an integral part of the global financial market, while substantial foreign capital has been entered China's capital market. [2]. By 2021, more than 420 billion funds had entered China's capital market, and will continue to increase further in the future. In this context, the economic effect of capital market liberalization has become a topic of critical concern in both academic and practical circles.

Investors use corporate environmental performance as a crucial indicator to understand companies' sustainable development [3],evaluate investment value [4], and reflect the corporate value in the stock price. Before liberalization, Chinese government has developed many

**Funding:** This study was supported by Philosophy and Social Science Project of Colleges and Universities in Jiangsu Province (2023SJYB2339). Suqian College Youth Research Fund Project (2023XQNA10).The funders had no role in study design, data collection and analysis, decision to publish, or preparation of the manuscript.

**Competing interests:** The authors have declared that no competing interests exist.

legal systems for environmental protection, but environmental problems remain serious. The reasons for the above problems are mainly manifested in the following aspects. On the one hand, some enterprises lack sufficient motivation to reduce the emission of pollutants facing pressure from the market, technology and capital. On the other hand some enterprises do not have enough knowledge about the environmental pollution problems, which leads to the ineffective solution. The liberalization of capital market provides an opportunity to solve the problem of environmental pollution. Some studies have demonstrated that capital market liberalization can improve corporate environmental performance [1, 5], but there is a lack of specificity in sample selection and the conclusions may lack applicability. This study examines the relationship between capital market liberalization and corporate environmental performance using heavily polluting firms as research samples. Results show that capital market liberalization significantly enhances corporate environmental performance. The mechanism of action test indicates that capital market liberalization enhances corporate environmental performance by enhancing investment in environmental assets, reducing stock volatility, and improving the quality of information disclosure. The heterogeneity test finds that capital market liberalization has significant effect on corporate environmental performance in Non-SOEs and firms with low internal control.

This investigation contributes to the literature on capital market liberalization in various ways. Firstly, this study examines the relation between capital market liberalization and environmental performance.Previous studies have also examined the relationship between market liberalization and environmental performance [1, 5]. However, their selection of samples and the quantification of environmental performance lacked specificity. This research is based on heavily polluting enterprises as the research samples, so the sample selection is more targeted. Besides, we use emission charges and expenditures on environmental taxes to quantify corporate environmental performance, so the research results will be closely related to its business-level environmental performance. Secondly, we discuss the effect of property rights and internal controls on the relationship between capital market liberalization and corporate environmental performance, which deepens the understanding of the impact mechanism of capital market literature on corporate environmental performance.Finally, We clarify the important pathways which capital market liberalization affects corporate environmental performance. Most of the literature analyzes the relationship between capital market liberalization and corporate environmental performance from the perspective of financing constraints [3] as well as corporate governance [7].Our findings confirm that capital market liberalization prompts heavily polluting firms to increase investment in environmental assets, reduce stock price volatility and improve the quality of disclosure, thereby improving environmental performance.

## 2. Literature review and hypothesis development

### 2.1 Literature review

**2.1.1 Capital market liberalization.** Existing literature on the economic consequences of capital market liberalization focuses mainly on the operation of capital markets,the behavior of enterprises, and the development of the real economy. First, capital market liberalization has introduced mature value investors to developing countries [1], which can guide investors to make value investments [6], improve the information content of stock prices [7], reduce the heterogeneous volatility of stock markets [8], and thus improve stock market stability and market effectiveness [9].Second, market transactions conducted by foreign investors can affect corporate decision-making behavior, such as improving the efficiency of corporate investment [1], reducing investment in inefficient labor [8],reducing commercial bank risk [10], reducing firms'earnings management [11],and promoting corporate social responsibility [12].Finally,

foreign investors can optimize corporate governance mechanisms to curb aggressive financial asset allocation by enterprises and enhance the quality of financial services to the real economy [13].

**2.1.2 Corporate environmental performance.**   Firstly, the factors influencing the corporate environmental performance. In relevant studies on corporate environmental performance, some studies are conducted at the macro level, Such as political connections [14], environmental protection tax [15], Climate risk [16], Low-carbon city pilot policy [17], and environmental regulation [18].Other studies focus on micro-level factors, Such as powerful CEOs [19], Multiple large shareholders [20];digital investment [21], corporate social responsibility [22],and genetic diversity on corporate boards [23]. Second, the importance of corporate environmental performance. Some scholars argue that green innovation (GI) can be used as an alternative to corporate environmental performance. Takalo et al.(2021) suggest that A large number of organizations and communities have been directed towards green innovation as a strategy to achieve environmental protection and economic growth [24]. Third, evaluating enterprises environmental performance. Tseng et al. (2013) assessed the green innovation practices of printed circuit board manufacturing companies in Taiwan in terms of four dimensions: management, process, product, and technological innovation, and also found that management innovation is a key driver of green innovation [25].

Scholars have conducted extensive research on capital market liberalization and environmental performance. In China, solving environmental problems relies mainly on government policies and regulations. The market has not played an adequate role in curbing the problem of environmental pollution. The liberalization of the capital market provides an opportunity to play the role of the market in combating the problem of environmental pollution. Therefore, we focus on the relationship between capital market liberalization and the environmental performance of heavily polluting enterprises.

## 2.2 Hypothesis development

We expect that capital market liberalization affects corporate environmental performance by increasing environmental assets, reducing stock price volatility, and improving the quality of disclosure.

First, capital market liberalization affects corporate environmental performance through investing environmental assets. Investments in environmental assets are characterized by large investments, long lead times and low short-term benefits. As the social effect brought about by environmental asset investment is greater than the economic benefit, this also leads to the fact that in enterprises with high financing constraints, environmental asset investment will reduce the funds for other project investments, causing enterprise costs to rise and profits to decrease. Opening a capital market effectively integrates China's originally closed market with the global market, enhancing the efficiency of China's capital market [25, 26]. Good environmental performance can attract new investors, and btain sufficient funds to alleviate financing constraints. Feng et al. (2022) find that capital market liberalization reduces firms' financing constraints, thereby enhancing corporate green innovation [27]. Heavy polluting enterprises have enough money to invest in environmental assets to achieve lower pollutant emissions and environmental taxes, which in turn improves corporate environmental performance.

Based on the above analysis, we propose the following hypothesis:

Hypothesis 1: Capital market liberalization enhances environmental performance by increasing investment in environmental assets.

Second, capital market liberalization leads to higher synchronization of China's capital market with the overseas capital market, which to a certain extent bears the impact of large-scale capital flows and thus affects capital market stability [27–29]. In particular, after international risk events, which include the U.S. subprime mortgage crisis and the 2015 stock market crash in China, substantial capital outflows from the capital market cause dramatic fluctuations in the market, increasing companies short-term financial risks. Environmental performance, as an important strategic resource, can alleviate the external market's impact on firms to some extent [30–32].In other words, enterprises with good environment- al performance can relieve stock price fluctuations to facilitate better access to funds for business growth through the capital markets [33–35].

Thus, firms should have the motivation and capability to enhance environmental performance.

Hypothesis 2: Capital market liberalization enhances environmental performance by reducing stock price volatility

Finally, Capital market liberalization may facilitate the enhancement of corporate environmental performance by improving the information environment. Kim and Zhang(2014) find that when firms take on more social responsibility, they can curb the spread of negative news by increasing the transparency of financial reporting [36].Nie et al.(2023) find that Capital market liberalization enhances corporate ESG disclosure and satisfies the needs of investors [37]. Thus, the pressure to disclose environmental information motivates heavily polluting firms to enhance their environmental protection efforts, thereby improving corporate environmental performance.

Based on the above analysis, we propose the following hypothesis:

Hypothesis 3: Capital market liberalization improves corporate environmental performance by improving the quality of disclosure.

## 3. Research design

### 3.1 Sample selection and data

The study uses difference-in-differences(DID) method to examine the impact of capital market liberalization on corporate environmental performance. We select China's A-share listed heavily polluting companies from 2010 to 2020 as the research sample, and perform the following on the initial sample:(1) Exclude the ST samples; and (2)Eliminate samples with missing main data. The heavy polluting companies referred to in this paper include thermal power, iron and steel, cement, electrolytic aluminum, coal, metallurgy, chemical, petrochemicals, building materials, paper making, brewing, pharmaceuticals, fermentation, textiles, tanning, and mining. We obtain 3248 firm-year observations. Financial data comes from CSMAR and the WIND database.

### 3.2 Variable definition

**3.2.1 Dependent variable.**   The dependent variable is corporate environmental performance. We use two methods to measure corporate environmental performance. The first method is measured using the ratio of the logarithm of emissions charges to the logarithm of operating revenues (EID).The EID indicator is positive. The second method uses the ratio of the logarithm of corporate environmental taxes to the logarithm of operating revenues (HEID).The HEID is an inverse indicator.

**3.2.2 Independent variable.** The independent variable is capital market liberalization (List*Post).List is a company dummy variable,which is 1 if the firm belong to the SH-HKSC, and 0 otherwise.Post is a time dummy variable, which is 1 when year is after 2014, and 0 otherwise. The definitions of the main variables are shown in Table 1.

## 3.3 Model design

The model to test the effect of capital market liberalization on corporate environmental performance:

$$HEID_{it}/EID_{it} = \beta_0 + \beta_1 List_i * Post_t + \sum controls + \gamma_t + \delta_i + \varepsilon_{it} \qquad (1)$$

HEID/EID represents Corporate environmental performance.List is a company dummy variable,which is 1 if the firm belong to the SH-HKSC, and 0 otherwise.Post is a time dummy variable, which is 1 when year is after 2014, and 0 otherwise.

# 4. Empirical analysis results

## 4.1 Descriptive statistics

Table 2 lists descriptive statistics results of each variable. From the main environmental performance variables, The minimum value of EID is 0.124, the maximum value is 0.425, and the standard deviation is 2.6870, which indicates that there are huge differences in the environmental performance of the sample firms. From the explaining variable,the average value of the List is 0.176, indicating that 17.6% of the sample companies entered the SH-HKSC. Other variables meet the expectations.

**Table 1. Definitions of related variables.**

| Variable Name | Variable Symbol | Variable Definition |
|---|---|---|
| **Corporate environmental performance** | EID | the ratio of the logarithm of emissions charges to the logarithm of operating revenues |
| | HEID | the ratio of the logarithm of corporate environmental taxes to the logarithm of operating revenues |
| **Underlying Stocks of Shanghai-Hong Kong Stock Connect** | List | List is a company dummy variable, which is 1 if the firm belong to the SH-HKSC, and 0 otherwise |
| **Shanghai-Hong Kong Stock Connect Launch Time** | Post | Post is a time dummy variable, which is 1 when year is after 2014, and 0 otherwise. |
| **Environmental assets** | EA | the ratio of environmental assets to total assets |
| **Stock price volatility** | VOL | the standard deviation of daily stock returns over the quarter |
| **Information disclosure quality** | DQ | DQ comes from CSMAR. The indicator is categorized into four levels and assigned values from 1 (bad) to 4 (good). |
| **Intangible assets ratio** | Intan | Intangible assets/all assets |
| **Financial leverage** | LEV | Total debt /total assets |
| **net profit growth rate** | Sg | (Ending net profit–beginning net profit)/ beginning net profit |
| **Independent director ratio** | DL | Proportion of the total number of independent directors on the board of directors |
| **Shareholding of major shareholders** | FS | Share ratio of major shareholders |
| **Institutional shareholding ratio** | INS | Shareholding ratio of institutional investors |

**Table 2. Descriptive statistics of the main variables.**

| Variable | N | Minimum | Maximum | Mean value | SD |
|---|---|---|---|---|---|
| EID | 3248 | 0.124 | 0.425 | 0.137 | 2.6870 |
| HEID | 3248 | 0.054 | 0.107 | 0.070 | 0.7021 |
| List | 3248 | 0.000 | 1.000 | 0.176 | 0.3050 |
| Post | 3248 | 0.000 | 1.000 | 0.365 | 0.4814 |
| Sg | 3248 | -0.586 | 3.320 | 0.225 | 0.5261 |
| Intan | 3248 | 0.034 | 0.357 | 0.164 | 0.0520 |
| LEV | 3248 | 0.048 | 0.932 | 0.435 | 0.2262 |
| DL | 3248 | 0.300 | 0.534 | 0.372 | 0.0522 |
| FS | 3248 | 0.084 | 0.746 | 0.358 | 15.208 |
| INS | 3248 | 0.000 | 0.624 | 0.324 | 0.4274 |

## 4.2 Basic regression

Table 3 shows the results of the regression between capital market liberalization and corporate environmental performance. Columns (1) and (2) do not control for year effects and firm effects, Columns (3) and (4) control for firm effects, and Columns (5) and (6) control the year and firm effects. In Columns (1),(3)and(5),The coefficients of List*Post are significantly positive.In Columns (2),(4)and(6),The coefficients of List*Post are significantly negative.The above regression results show that, compared with firms that have not been listed in the SH-HKSC, the environmental performance reflected by the SH-HKSC firms has remarkably improved, also proving that capital market liberalization promotes corporate environmental performance.

## 4.3 Robustness test

**4.3.1 Parallel trend test.** Parallel trend testing is necessary to ensure the validity of the DID model. Following Jia et al.(2022) [38], the year 2014 when SH-HKSC policy was implemented is taken as the benchmark, and 10 years of dummy variables are set. If the year is after

**Table 3. Capital market liberalization and corporate environmental performance.**

| Variables | EID(1) | HEID(2) | EID(3) | HEID(4) | EID(5) | HEID(6) |
|---|---|---|---|---|---|---|
| List*Post | 0.052** (6.74) | -0.065*** (-8.14) | 0.056*** (8.20) | -0.055** (-6.05) | 0.060*** (4.62) | -0.062*** (-7.51) |
| Size | 0.021** (3.64) | 0.022** (2.99) | 0.038* (2.05) | 0.034** (3.21) | 0.020* (2.13) | 0.030** (3.57) |
| Intan | 0.002 (0.77) | 0.001 (0.72) | 0.001 (0.30) | 0.002 (0.12) | 0.001 (0.15) | 0.001 (0.20) |
| LEV | -0.033** (-2.84) | -0.032* (-2.03) | -0.024* (-1.92) | -0.025** (-3.05) | -0.041* (-1.62) | -0.038* (1.97) |
| DL | 0.042** (3.24) | 0.039* (2.49) | 0.052** (3.67) | 0.055* (2.29) | 0.063** (4.09) | 0.059*** (6.92) |
| FS | -0.008** (-1.43) | -0.007* (-1.62) | -0.005** (-2.06) | -0.004** (-1.92) | -0.006** (-2.25) | -0.005* (-1.52) |
| INS | 0.018*** (4.92) | 0.019*** (4.09) | 0.020*** (4.38) | 0.017*** (5.63) | 0.013*** (4.95) | 0.015** (2.37) |
| Constant | -1.528*** (-22.65) | 2.524*** (45.36) | -2.124*** (-30.21) | 1.021*** (16.38) | 3.021*** (62.35) | -1.348*** (-25.31) |
| Year Fixed effect | NO | NO | NO | NO | YES | YES |
| Firm Fixed effect | NO | NO | YES | YES | YES | YES |
| R² | 0.142 | 0.145 | 0.135 | 0.129 | 0.121 | 0.136 |
| N | 3248 | 3248 | 3248 | 3248 | 3248 | 3248 |

T-statistics for the regression coefficients are in parentheses.

*, **, and *** represent statistical significance at the 10%, 5%, and 1% levels, respectively.

**Table 4. Hypothesis test for parallel trend.**

| Variable | EID | HEID |
|---|---|---|
| List_2010 | 0.015(0.36) | 0.025(0.65) |
| List_2011 | 0.052(0.11) | 0.033(0.75) |
| List_2012 | 0.094(1.62) | 0.062(1.72) |
| List_2013 | 0.125(0.96) | 0.0547(0.22) |
| List_2015 | 0.129**(3.28) | -0.036***(-6.35) |
| List_2016 | 0.132***(5.27) | -0.257***(-6.97) |
| List_2017 | 0.136***(7.21) | -0.254***(-6.30) |
| List_2018 | 0.085**(3.29) | -0.206***(-5.61) |
| List_2019 | 0.124*(1.35) | -0.096**(-3.05) |
| List_2020 | 0.024**(3.62) | -0.217***(-6.27) |
| Constant | 0.308***(8.30) | -0.105**(-3.06) |
| Controls | YES | YES |
| Year fixed effect | YES | YES |
| Firm fixed effect | YES | YES |
| $R^2$ | 0.141 | 0.112 |
| N | 3248 | 3248 |

T-statistics for the regression coefficients are in parentheses.

*, **, and *** represent statistical significance at the 10%, 5%, and 1% levels, respectively.

2014, current is 1,otherwise it is 0. They were cross-multiplied in Models 1 to List.The results are shown in Table 4. It can be seen that the regression coefficients of List_2010-List_2013 are insignificant,and the regression coefficients of List_2015- List_2020 are significantly,which fully indicates that corporate environmental performance has indeed been significantly improved after the implementation of the SH-HKSC, also proving that the DID model constructed is effective.

**4.3.2. PSM-DID test.** The underlying stocks of the SH-HKSC are artificially selected by the China Securities Regulatory Commission. The general principle of selection is that companies with good business performance, less risk, and good governance, which may have the problem of self-selection. Hence, the PSM1:1 nearest neighbour approach is adopted for matching, that is, searching for matching samples among the underlying stocks of the SH-HKSC. Specifically, the market value, return on equity, enterprise growth rate, and year are included as indicators of the control samples, which are the companies with the closest scores for the matching variables, and 4271 total samples are obtained. The estimated coefficients shown in Columns (1) and (2) of Table 5 are consistently significantly correlated, indicating that the implementation of the SH-HKSC policy can significantly enhance corporate environmental performance.

**4.3.3 Main variable alternative.** Re-quantifying the variables is the main way to conduct robustness tests.Thus, we replace the explained variable with ratio of environmental fines to operating revenue in Column (1) of Table 6.We expect that the amount of environmental fines will be reduced after the implementation of the SH-HKSC. The estimated coefficient of List*-Post is significantly negative, indicating that the results of this study are robust.

**4.3.4 Sample adjustment.** Capital market liberalization enhances corporate environmental performance. However, for A + H cross-listed and QFII holding companies, the dual supervision by both China and Hong Kong has driven better environmental performance. To prevent interference, we exclude firms listed on both the a-share and h-share exchanges as well

**Table 5. PSM-DID test.**

| Variable | EID (1) | HEID (2) |
|---|---|---|
| List*Post | 0.205***(5.62) | -0.152***(-4.74) |
| Constant | 10.319***(23.62) | 2.052***(12.35) |
| Controls | YES | YES |
| Year Fixed effect | YES | YES |
| Firm Fixed effect | YES | YES |
| $R^2$ | 0.120 | 0.134 |
| N | 4271 | 4271 |

T-statistics for the regression coefficients are in parentheses.

*, **, and *** represent statistical significance at the 10%, 5%, and 1% levels, respectively.

as QFII holdings. The estimated coefficients shown in Columns (2) and (3) of Table 6 are consistently significantly positive, indicating that capital market liberalization enhances corporate environmental performance indicating the robustness of the conclusion.

**4.3.5 Placebo test.** In order to verify that the improvement in environmental performance of heavy polluting enterprises is due to the implementation of and not due to other factors. We advance the implementation of the SH-HKSC policy by two years, and if the coefficient of List*Post is significant, indicating that the improvement of environmental performance of heavy polluters is not related to the opening of the capital market.The estimated coefficients of List*Post shown in Columns (1) and (2) of Table 7 are insignificant, indicating that indicating that the improvement in environmental performance of heavy polluting enterprises is due to capital market liberalization, thus validating the conclusions of this study.

**4.3.6 Heckman test.** We use the Heckman two-stage model to control for the problem of self-selection of samples. Referring to Zhang (2023),In addition to the original control variables, we add variables that may affect the selection of firms into the SH-HKSC, including firm value (TobinQ), Listage (Age), stock turnover(Turnover), and dividend payout ratio (Dividen). The first stage probit regression results are shown in column 1 of Table 8. The results of Heckman's second stage regression are in Column 2 and 3 of Table 8, and the inverse Mills coefficient (imr) is significant at the 1 per cent level. The regression coefficients for List*Post remain significant when we control for relevant control variables, time effects, and firm effects, consistent with the baseline regression results.

**Table 6. Robustness test.**

| Variable | PUB(1) | EID(2) | HEID(3) |
|---|---|---|---|
| List*Post | -1.32***(-6.57) | 0.103**(6.39) | -0.092*(-5.24) |
| Constant | 3.062***(20.35) | -1.057***(-10.00) | 0.327***(12.02) |
| Controls | YES | YES | YES |
| Year fixed effect | YES | YES | YES |
| Firm fixed effect | YES | YES | YES |
| $R^2$ | 0.135 | 0.280 | 0.262 |
| N | 3248 | 2783 | 2783 |

T-statistics for the regression coefficients are in parentheses.

*, **, and *** represent statistical significance at the 10%, 5%, and 1% levels,respectively.

**Table 7. Placebo test.**

| Variable | EID(1) | HEID(2) |
|---|---|---|
| List*Post | -0.039(-0.36) | 0.015(1.30) |
| Constant | 1.020***(6.21) | 0.524***(11.25) |
| Controls | YES | YES |
| Year fixed effect | YES | YES |
| Firm fixed effect | YES | YES |
| $R^2$ | 0.206 | 0.247 |
| N | 3248 | 3248 |

statistics for the regression coefficients are in parentheses.

*, **, and *** represent statistical significance at the 10%, 5%, and 1% levels,respectively.

**4.4 Heterogeneity analysis.** Table 9 shows the regression results of the firm heterogeneity. We categorize firms into state-owned and non-state-owned firms(SOEs and Non-SOEs), and firms with high internal controls and low internal controls. The List*Post coefficients are significantly higher for Non-SOEs and firms with low internal control than for SOEs and firms with high internal control, respectively. Firstly, compared to SOEs, non-SOEs have more incentives to illustrate the effectiveness of their environmental protection through good environmental performance in order to gain government policy support and investors' attention. Secondly, firms with low internal control are more willing to obtain investment from overseas investors through good environmental performance under the pressure of capital market liberalisation compared to firms with high internal control.

**Table 8. Heckman test.**

| Variable | SH-HKSC(1) | EID(2) | HEID(3) |
|---|---|---|---|
| List*Post | | 0.060***(11.05) | -0.059***(-6.94) |
| Size | 0.025*(1.20) | 0.023**(3.06) | 0.019*(2.64) |
| Sg | 0.126**(4.28) | 0.102*(2.65) | 0.096*(4.36) |
| Intan | 0.001(0.10) | 0.001(0.00) | 0.002(0.00) |
| LEV | -0.032*(1.34) | -0.114**(6.95) | -0.157**(11.03) |
| DL | 0.241***(11.52) | 0.138***(16.30) | 0.206***(15.30) |
| FS | -0.057**(-5.03) | -0.048*(-2.92) | -0.062*(-3.61) |
| INS | 0.359***(22.03) | 0.267***(19.63) | 0.168***(16.30) |
| TobinQ | 0.143***(9.06) | | |
| Age | 0.010*(3.03) | | |
| Turnover | 0.031(1.37) | | |
| Dividen | 0.548***(33.64) | | |
| imr | | 1.305***(36.02) | 0.962***(30.87) |
| Year fixed effect | NO | YES | YES |
| Firm fixed effect | NO | YES | YES |
| $R^2$ | 0.103 | 0.268 | 0.251 |
| N | 3248 | 3248 | 3248 |

T-statistics for the regression coefficients are in parentheses.

*, **, and *** represent statistical significance at the 10%, 5%, and 1% levels,respectively.

**Table 9. Heterogeneity analysis.**

| Variable | SOEs | | Non-SOEs | | high internal controls | | low internal controls | |
|---|---|---|---|---|---|---|---|---|
| | EID | HEID | EID | HEID | EID | HEID | EID | HEID |
| List*Post | 0.017(1.34) | -0.015(-1.52) | 0.247***(6.84) | -0.354***(-5.25) | 0.063(2.02) | -0.072(-2.58) | 0.098**(4.89) | -0.085**(-5.07) |
| Constant | 2.359***(6.32) | 1.018***(2.06) | 0.102*(2.03) | 1.029***(15.30) | 5.021***(5.02) | -2.014***(-1.35) | 9.024***(6.27) | 2.359***(12.05) |
| Controls | YES | YES | YES | YES | YES | YES | YES | YES |
| Year fixed effect | YES | YES | YES | YES | YES | YES | YES | YES |
| Firm fixed effect | YES | YES | YES | YES | YES | YES | YES | YES |
| $R^2$ | 0.122 | 0.201 | 0.251 | 0.327 | 0.120 | 0.228 | 0.239 | 0.105 |
| N | 2087 | 2087 | 1161 | 1161 | 1401 | 1401 | 1847 | 1847 |
| Chow test | P<0.1 | | | | P<0.1 | | | |

T-statistics for the regression coefficients are in parentheses.

*, **, and *** represent statistical significance at the 10%, 5%, and 1% levels, respectively.

## 4.5 Mediation analysis

The above studies have confirmed that capital market liberalization enhances the environmental performance of heavy polluting enterprises, but the mechanism of action has not been clarified. Thus, we use investment in environmental assets, stock price volatility, and disclosure quality as mediating variables and utilize a three-step approach to test the mediating effect (Feng et.al,2022) [24].The specific modeling is as follows:

$$Moderator_{it} = \beta_0 + \beta_1 List_i * Post_t + \sum Controls + \gamma_t + \delta_i + \varepsilon_{it} \qquad (2)$$

$$HEID_{it}/EID_{it} = \beta_0 + \beta_1 List_i * Post_t + \beta_2 Moderator_{it} + \sum Controls + \gamma_t + \delta_i + \varepsilon_{it} \qquad (3)$$

**4.5.1 Environmental assets (EA).** The results of the mediating effects of environmental assets are presented in Table 10. As can be seen,the estimated coefficient of List*Post in column (2) is significantly positive,which indicates that capital market liberalization prompts heavily polluting firms to increase investment in environmental assets. When List*Post and EA are regressed together as environmental performance explanatory variables, the regression coefficient of List*Post decreases to 0.035 compared to column (1), while the regression

**Table 10. Mediation effect results(environmental assets).**

| Variable | EID (1) | EA(2) | EID(3) |
|---|---|---|---|
| List*Post | 0.060***(4.62) | 0.058**((3.05) | 0.035**(2.54) |
| EA | | | 0.067**((2.07) |
| Constant | 3.021**((62.35) | 1.041**(9.62) | 0.921**(11.02) |
| Controls | YES | YES | YES |
| Year fixed effect | YES | YES | YES |
| Firm fixed effect | YES | YES | YES |
| $R^2$ | 0.121 | 0.134 | 0.159 |
| N | 3248 | 3248 | 3248 |

T-statistics for the regression coefficients are in parentheses.

*, **, and *** represent statistical significance at the 10%, 5%, and 1% levels, respectively.

**Table 11. Mediation effect results(Stock price volatility).**

| Variable | EID (1) | VOL(2) | EID(3) |
|---|---|---|---|
| List*Post | 0.060***(4.62) | -0.096**(-6.62) | 0.052***(4.30) |
| VOL | | | -0.081**(-6.31) |
| Constant | 3.021***(62.35) | 1.214***(12.20) | 2.010***(32.50) |
| Controls | YES | YES | YES |
| Year fixed effect | YES | YES | YES |
| Firm fixed effect | YES | YES | YES |
| $R^2$ | 0.121 | 0.134 | 0.159 |
| N | 3248 | 3248 | 3248 |

T-statistics for the regression coefficients are in parentheses.

*, **, and *** represent statistical significance at the 10%, 5%, and 1% levels,respectively.

coefficient of EA remains significant. Based on this calculations, the mediating effect of EA is 46.7%.The above analysis suggests that the mediating effect of environmental assets holds. It indicates that capital market liberalization pushes heavily polluting enterprises to increase their investment in environmental assets, which in turn leads to an improvement in corporate environmental performance.

**4.5.2 Stock price volatility(VOL).** The results of the mediating effects of stock price volatility are presented in Table 11.As can be seen, the estimated coefficient of List*Post in column (2) is significantly negative, which indicates that capital market liberalization significantly reduces stock price volatility.When List*Post and VOL are regressed together as environmental performance explanatory variables, the regression coefficient of List*Post decreases to 0.052 compared to column (1), while the regression coefficient of VOL remains significant. Based on this calculations, the mediating effect of VOL is 13.3%.This suggests that VOL mediates the effect of capital market liberalization on environmental performance. It indicates that capital market liberalization exacerbates the impact of risky events on stock prices, while good environmental performance can effectively alleviate this impact. This also provides firms with access to stable funding from the capital market, which contributes to their environmental performance.

**4.5.3 Information disclosure quality (DQ).** We also report results for the mediating effect of disclosure quality in Table 12.The regression coefficient of List*Post is significantly positive in columns (2), which indicates that capital market liberalization prompts heavily polluting firms to improve information disclosure quality. When List*Post and IDQ are regressed together as environmental performance explanatory variables, the coefficient of List*Post decreases to 0.023 compared to column (1), while the regression coefficient of DQ remains significant. Based on this calculations, the mediating effect of DQ is 61.7%.This suggests that DQ mediates the effect of capital market liberalization on environmental performance. It indicates that capital market liberalization enhances the quality of information disclosure by heavy polluters, reduces the degree of information asymmetry between corporate management and investors, and alleviates the financing constraints due to information asymmetry thereby promoting environmental performance.

## 5. Conclusion

Capital market liberalization is an inevitable choice for economic development and enhances China's high-quality economic growth. In the context of SH-HKSC, We select 3248 samples of

**Table 12. Mediation effect results (Information disclosure quality).**

| Variable | EID (1) | DQ(2) | EID(3) |
|---|---|---|---|
| List*Post | 0.060***(4.62) | 0.152***(5.03) | 0.023***(2.08) |
| DQ | | | 0.241***(4.14) |
| Constant | 3.021***(62.35) | 0.814***(11.20) | 1.025***(9.61) |
| Controls | YES | YES | YES |
| Year fixed effect | YES | YES | YES |
| Firm fixed effect | YES | YES | YES |
| $R^2$ | 0.121 | 0.134 | 0.204 |
| N | 3248 | 3248 | 3248 |

T-statistics for the regression coefficients are in parentheses.

*, **, and *** represent statistical significance at the 10%, 5%, and 1% levels,respectively.

heavily polluting firms listed in China in 2010–2020 to investigate the impact of capital market liberalization on environmental performance. We find that capital market liberalization significantly enhance corporate environmental performance.

The effect varies across ownership types as well as across firms with different internal controls. Finally, we also find that capital market liberalization promotes corporate environmental performance through increasing environmental assets as well as improving the quality of information disclosure.

This study provides new evidence on the economic effects of capital market liberalization and expands the research related to corporate environmental performance.For Governments, Drawing on the experience of the smooth implementation of the SH-HKSC, the government should continue to promote the opening of the capital market in a prudent and reasonable manner, gradually expand and deepen the level of opening of the capital market, accelerate the docking of the capital market with the international capital market [5], draw on the capital market systems of international developed countries and regions, and actively draw on their experience in order to establish and improve the capital market system, and actively make use of the international capital, talents and systems to provide a good and effective capital market environment for the green development of enterprises and even for the green transformation of the economy and society. For enterprises, it is important to take advantage of the opportunities arising from the opening of the capital market to raise awareness of corporate social responsibility and to achieve sustainable development by improving environmental performance.

## Acknowledgments

To the periodical office and two anonymous reviewers for their help towards improving this work.

## Author Contributions

**Writing – original draft:** Qi'an Zhong.

**Writing – review & editing:** Qi'an Zhong.

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
