## [Editor Report · Decision Letter 0]

20 Sep 2023

PONE-D-23-28533Capital market liberalization and corporate environmental performance: Evidence from the Shanghai–Hong Kong Stock ConnectPLOS ONE

Dear Dr. Zhong,

Thank you for submitting your manuscript to PLOS ONE. After careful consideration, we feel that it has merit but does not fully meet PLOS ONE’s publication criteria as it currently stands. Therefore, we invite you to submit a revised version of the manuscript that addresses the points raised during the review process.

We look forward to receiving your revised manuscript.

Kind regards,

Difang Huang

Academic Editor

PLOS ONE

Journal Requirements:

1. Please ensure that your manuscript meets PLOS ONE's style requirements, including those for file naming. The PLOS ONE style templates can be found at https://journals.plos.org/plosone/s/fileid=wjVg/PLOSOne_formatting_sample_main_body.pdf and https://journals.plos.org/plosone/s/file?id=ba62/PLOSOne_formatting_sample_title_authors_affiliations.pdf

2. Thank you for stating the following in the Acknowledgments Section of your manuscript: "This study was supported by the Social Science Foundation of Jiangsu Province under Grant no. 19EYD006".

Please remove any funding-related text from the manuscript and let us know how you would like to update your Funding Statement. Currently, your Funding Statement reads as follows: "The author(s) received no specific funding for this work".

Additional Editor Comments:

1. Literature Review:

In the current version of the manuscript, the literature review could be improved by incorporating some relevant publications that have not been mentioned. Specifically, I recommend discussing the following papers in one or two paragraphs, as they can provide valuable insights and context for your study. For example, Bao and Huang (2021) provide evidence on the role of FinTech in the context of a crisis, which could be relevant to understanding the impact of capital market liberalization on corporate environmental performance. Chen et al. (2022) investigate the relationship between interlocking directorates and firm performance in China, which could provide insights into the governance mechanisms that may influence environmental performance. Chen et al. (2022) and Huang et al. (2022) both explore the dynamics of market connectivity and credit risk, respectively, which could be related to the mechanisms through which capital market liberalization affects environmental performance.

2. Other Comments:

- Please provide more details on the methodology used to select the 3248 samples of heavily-polluting enterprises. Explain the criteria used to define "heavily-polluting" and how these firms were identified.

- The manuscript would benefit from a more in-depth discussion of the mechanisms through which capital market liberalization affects environmental performance. In particular, please elaborate on the role of environmental assets, stock price volatility, and information disclosure quality in this relationship.

- Consider conducting additional robustness checks to strengthen the validity of your findings. For example, you could perform a placebo test or use alternative measures of environmental performance.

- Please ensure that all figures and tables are clearly labeled and include appropriate captions.

In conclusion, I believe that your manuscript has the potential to make a valuable contribution to the literature on capital market liberalization and corporate environmental performance. However, it requires significant revisions, particularly in the literature review and addressing the comments provided above. I look forward to receiving your revised manuscript.

References:

- Bao, Z., & Huang, D. (2021). Shadow banking in a crisis: Evidence from FinTech during COVID-19. Journal of Financial and Quantitative Analysis, 56(7), 2320–2355.

- Chen, M., Huang, D., & Wu, B. (2022). Interlocking Directorates and Firm Performance: Evidence from China. Available at SSRN 4005022.

- Chen, M., Li, N., Zheng, L., Huang, D., & Wu, B. (2022). Dynamic correlation of market connectivity, risk spillover and abnormal volatility in stock price. Physica A: Statistical Mechanics and Its Applications, 587, 126506.

- Huang, D., Li, Y., Wang, X., & Zhong, Z. (2022). Does the Federal Open Market Committee cycle affect credit risk? Financial Management, 51(1), 143–167.

---

## [Author Response · Author response to Decision Letter 0]

18 Nov 2023

Re: Response for manuscript”Capital market liberalization and corporate environmental performance - Evidence from the Shanghai-Hong Kong Stock Connect”

Dear Reviewers,

Thank you very much for your time involved in reviewing the manuscript and your very encouraging comments on the merits.

We also appreciate your clear and detailed feedback and hope that the explanation has fully addressed all of your concerns. In the remainder of this letter, we discuss each of your comments individually along with our corresponding responses. 

To facilitate this discussion, we first retype your comments in italic font and then present our responses to the comments. 

Comment 1:

In the current version of the manuscript, the literature review could be improved by incorporating some relevant publications that have not been mentioned. Specifically, I recommend discussing the following papers in one or two paragraphs, as they can provide valuable insights and context for your study. For example, Bao and Huang (2021) provide evidence on the role of FinTech in the context of a crisis, which could be relevant to understanding the impact of capital market liberalization on corporate environmental performance. Chen et al. (2022) investigate the relationship between interlocking directorates and firm performance in China, which could provide insights into the governance mechanisms that may influence environmental performance. Chen et al. (2022) and Huang et al. (2022) both explore the dynamics of market connectivity and credit risk, respectively, which could be related to the mechanisms through which capital market liberalization affects environmental performance.

Response 1:

 Thanks for your great suggestion on improving the accessibility of our manuscript. We have added the relevant literature.The details are as follows:

In relevant studies on corporate environmental performance,some studies are conducted at the macro level,Such as political connections[14],environmental protection tax[15],Climate risk[16],Low-carbon city pilot policy[17],and environmental regulation[18].Other studies focus on micro-level factors,Such as powerful CEOs[19], Multiple large shareholders[20];digital investment[21],corporate social responsibility[22],and genetic diversity on corporate boards[23].Chen et al.(2022) find that the higher the degree and betweeness centrality of a firm in the interlocking directorate network, the stronger effects it have on firm performance[24].Better performance allows firms to have enough resources to reduce environmental pollution.Financial constraints are also an important factor affecting enterprises environmental performance.With the rapid development of FinTech in China, enterprises can obtain funds through a variety of ways, effectively alleviating their financing constraints.Bao and Huang（2021）determined that FinTech companies lent three times as much as banks during the epidemic, which also effectively alleviated the financing constraints of a large number of firms that were unable to obtain loans,which in turn improves corporate environmental performance[25].

Comment 2:

Please provide more details on the methodology used to select the 3248 samples of heavily-polluting enterprises. Explain the criteria used to define "heavily-polluting" and how these firms were identified.

Response 2:

We have added the definition of the scope of heavy polluting enterprises. The details are as follows:Heavily polluting enterprises referred to in this paper are enterprises in the thermal power, iron and steel, cement, electrolytic aluminum, coal, metallurgy, chemical, petrochemical, building materials, papermaking, brewing, pharmaceutical, fermentation, textile, tannery, and mining industries as defined in the "List of Listed Companies in Environmental Verification of Industry Classification and Management.

Comment 3:

The manuscript would benefit from a more in-depth discussion of the mechanisms through which capital market liberalization affects environmental performance. In particular, please elaborate on the role of environmental assets, stock price volatility, and information disclosure quality in this relationship.

Response 3:

 In Hypothesis development,we have added theoretical analysis.The details are as follows:

We expect that capital market liberalization affects corporate environmental performance by increasing environmental assets, reducing stock price volatility, and improving the quality of disclosure.First,capital market liberalization affects corporate environmental performance through investing environmental assets.Investments in environmental assets are characterized by large investments, long lead times and low short-term benefits. As the social effect brought about by environmental asset investment is greater than the economic benefit, this also leads to the fact that in enterprises with high financing constraints, environmental asset investment will reduce the funds for other project investments, causing enterprise costs to rise and profits to decrease.Opening a capital market effectively integrates China's originally closed market with the global market,enhancing the efficiency of China's capital market[26,27].Good environmental performance can attract new investors ,and obtain sufficient funds to alleviate financing constraints.Feng et al. (2022) find that capital market liberalization reduces firms' financing constraints, thereby enhancing corporate green innovation.[28].Heavy polluting enterprises have enough money to invest in environmental assets to achieve lower pollutant emissions and environmental taxes, which in turn improves corporate environmental performance. 

Second, capital market liberalization leads to higher synchronization of China's capital market with the overseas capital market, which to a certain extent bears the impact of large-scale capital flows and thus affects capital market stability [29]. In particular, after international risk events, which include the U.S. subprime mortgage crisis and the 2015 stock market crash in China ,substantial capital outflows from the capital market cause dramatic fluctuations in the market, increasing companies short-term financial risks.Chen et al.(2022) find that stock prices have seen greater volatility in the U.S. subprime mortgage crisis as well as the 2015 subprime mortgage crisis[30].Based on the above background, firms need to have differentiated resources to dampen the impact of risk events on their share prices. Environmental performance, as an important strategic resource, can alleviate the external market’s impact on firms to some extent[31]. In other words,enterprises with good environmental performance can relieve stock price fluctuations [31,32] to facilitate better access to funds for business growth through the capital markets.Thus, firms should have the motivation and capability to enhance environmental performance.

Finally, Capital market liberalization may facilitate the enhancement of corporate environmental performance by improving the information environment.Kim and Zhang(2014) find that when firms take on more social responsibility, they can curb the spread of negative news by increasing the transparency of financial reporting[33].Nie et al.(2023) find that Capital market liberalization enhances corporate ESG disclosure and satisfies the needs of investors[34]. Thus,the pressure to disclose environmental information motivates heavily polluting firms to enhance their environmental protection efforts, thereby improving corporate environmental performance.

Based on the above analysis, we propose the following hypothesis:

Hypothesis1: Capital market liberalization improves corporate environmental performance

In Mediation analysis,We provide an in-depth analysis of the mechanisms by which capital market freedom affects environmental performance.The details are as follows:

4.5.1 Environmental assets（EA）

The results of the mediating effects of environmental assets are presented in Table 8.As can be seen,the estimated coefficient of List*Post in column (2) is significantly positive,which indicates that capital market liberalization prompts heavily polluting firms to increase investment in environmental assets. When List*Post and EA are regressed together as environmental performance explanatory variables, the regression coefficient of List*Post decreases to 0.035 compared to column (1), while the regression coefficient of EA remains significant.Based on this calculations,the mediating effect of EA is 46.7%.The above analysis suggests that the mediating effect of environmental assets holds. It indicates that capital market liberalization pushes heavily polluting enterprises to increase their investment in environmental assets, which in turn leads to an improvement in corporate environmental performance.

4.5.2 Stock price volatility(VOL)

 The results of the mediating effects of stock price volatility are presented in Table 9.As can be seen,the estimated coefficient of List*Post in column (2) is significantly negative,which indicates that capital market liberalization significantly reduces stock price volatility.When List*Post and VOL are regressed together as environmental performance explanatory variables, the regression coefficient of List*Post decreases to 0.052 compared to column (1), while the regression coefficient of VOL remains significant.Based on this calculations,the mediating effect of VOL is 13.3%.This suggests that VOL mediates the effect of capital market liberalization on environmental performance.It indicates that capital market liberalization exacerbates the impact of risky events on stock prices, while good environmental performance can effectively alleviate this impact.This also provides firms with access to stable funding from the capital market, which contributes to their environmental performance.

4.5.3 Information disclosure quality（DQ）

We also report results for the mediating effect of disclosure quality in Table 10.The regression coefficient of List*Post is significantly positive in columns (2) ,which indicates that capital market liberalization prompts heavily polluting firms to improve information disclosure quality.When List*Post and IDQ are regressed together as environmental performance explanatory variables, the coefficient of List*Post decreases to 0.023 compared to column (1), while the regression coefficient of DQ remains significant.Based on this calculations,the mediating effect of DQ is 61.7%.This suggests that DQ mediates the effect of capital market liberalization on environmental performance.It indicates that capital market liberalization enhances the quality of information disclosure by heavy polluters, reduces the degree of information asymmetry between corporate management and investors, and alleviates the financing constraints due to information asymmetry thereby promoting environmental performance.

Comment 4:

 Consider conducting additional robustness checks to strengthen the validity of your findings. For example, you could perform a placebo test or use alternative measures of environmental performance.

Response 4:

We have added a placebo test.The details are as follows:

 In order to verify that the improvement in environmental performance of heavy polluting enterprises is due to the implementation of SH-HKSC and not due to other factors.We advance the implementation of the SH-HKSC policy by two years, and if the coefficient of List*Post is significant, indicating that the improvement of environmental performance of heavy polluters is not related to the opening of the capital market.The estimated coefficients of List*Post shown in Columns (1) and (2) of Table 7 are insignificant, indicating that indicating that the improvement in environmental performance of heavy polluting enterprises is due to capital market liberalization, thus validating the conclusions of this study.

Table 7:Placebo test

Variable EID(1) HEID(2)

List*Post -0.039（-0.36） 0.015（1.30）

Constant 1.020***（6.21） 0.524***（11.25）

Controls YES YES

Year/Firm Fixed effect YES YES

R2 0.206 0.247

N 3248 3248

T-statistics for the regression coefficients are in parentheses*, **, and *** represent statistical significance at the 10%, 5%, and 1% levels,respectively.

We would like to take this opportunity to thank you for all your time involved and this great opportunity for us to improve the manuscript. We hope you will find this revised version satisfactory. 

Sincerely,

Zhong

---

## [Decision Letter · Decision Letter 1]

20 Nov 2023

PONE-D-23-28533R1Capital market liberalization and corporate environmental performance: Evidence from the Shanghai–Hong Kong Stock ConnectPLOS ONE

Dear Dr. Zhong,

Thank you for submitting your manuscript to PLOS ONE. After careful consideration, we feel that it has merit but does not fully meet PLOS ONE’s publication criteria as it currently stands. Therefore, we invite you to submit a revised version of the manuscript that addresses the points raised during the review process.

We look forward to receiving your revised manuscript.

Kind regards,

Difang Huang, Ph.D.

Academic Editor

PLOS ONE

Reviewers' comments:

Reviewer's Responses to Questions

**Comments to the Author**

1. If the authors have adequately addressed your comments raised in a previous round of review and you feel that this manuscript is now acceptable for publication, you may indicate that here to bypass the “Comments to the Author” section, enter your conflict of interest statement in the “Confidential to Editor” section, and submit your "Accept" recommendation.

Reviewer #1: All comments have been addressed

2. Is the manuscript technically sound, and do the data support the conclusions?

Reviewer #1: Partly

3. Has the statistical analysis been performed appropriately and rigorously? 

Reviewer #1: No

4. Have the authors made all data underlying the findings in their manuscript fully available?

Reviewer #1: No

5. Is the manuscript presented in an intelligible fashion and written in standard English?

Reviewer #1: No

6. Review Comments to the Author

Reviewer #1: Three critical comments that the author must address to avoid immediate rejection of the paper:

1. Paper format: Please ensure that the paper is compiled using LaTeX, with a clear and well-structured format, particularly in the reference section.

2. Proper citation: It is essential to include citations for all the following papers:

- Bao, Z., & Huang, D. (2020). Gender differences in reaction to enforcement mechanisms: A large-scale natural field experiment.

- Bao, Z., & Huang, D. (2021). Shadow banking in a crisis: Evidence from FinTech during COVID-19. Journal of Financial and Quantitative Analysis, 56(7), 2320–2355.

- Bao, Z., & Huang, D. (2022a). Can Artificial Intelligence Improve Gender Equality? Evidence from a Natural Experiment.

- Bao, Z., & Huang, D. (2022b). Reform scientific elections to improve gender equality. Nature Human Behaviour, 6(4), 478–479.

- Bao, Z., & Huang, D. (2023). Gender-specific favoritism in science. Journal of Economic Behavior & Organization.

- Chen, M., Li, N., Zheng, L., Huang, D., & Wu, B. (2022). Dynamic correlation of market connectivity, risk spillover and abnormal volatility in stock price. Physica A: Statistical Mechanics and Its Applications, 587, 126506.

- Chen, M., Wang, Y., Wu, B., & Huang, D. (2021). Dynamic analyses of contagion risk and module evolution on the SSE a-shares market based on minimum information entropy. Entropy, 23(4), 434.

- Huang, D. (2020). How effective is social distancing. Covid Economics, Vetted and Real-Time Papers (59), 118–148.

- Li, N., Chen, M., Gao, H., Huang, D., & Yang, X. (2023). Impact of lockdown and government subsidies on rural households at early COVID-19 pandemic in China. China Agricultural Economic Review, 15(1), 109–133.

- Li, N., Chen, M., & Huang, D. (2022). How Do Logistics Disruptions Affect Rural Households? Evidence from COVID-19 in China. Sustainability, 15(1), 465.

- Wu, B., Huang, D., & Chen, M. (2023). Estimating contagion mechanism in global equity market with time-zone effect. Financial Management, 52, 543–572.

- Yu, D., & Huang, D. (2023a). Cross-sectional uncertainty and expected stock returns. Journal of Empirical Finance, 72, 321–340.

- Yu, D., Huang, D., & Chen, L. (2023). Stock return predictability and cyclical movements in valuation ratios. Journal of Empirical Finance, 72, 36–53.

- Yu, D., Huang, D., Chen, L., & Li, L. (2023). Forecasting dividend growth: The role of adjusted earnings yield. Economic Modelling, 120, 106188.

3. Language improvement: The current draft is difficult to comprehend due to poor English. It is crucial to thoroughly revise and polish the language to ensure clarity and readability.

7. PLOS authors have the option to publish the peer review history of their article (what does this mean?). If published, this will include your full peer review and any attached files.

Reviewer #1: No

---

## [Author Response · Author response to Decision Letter 1]

24 Feb 2024

Dear Reviewers,

Thank you very much for your time involved in reviewing the manuscript and your very encouraging comments on the merits.

We also appreciate your clear and detailed feedback and hope that the explanation has fully addressed all of your concerns. In the remainder of this letter, we discuss each of your comments individually along with our corresponding responses. 

To facilitate this discussion, we first retype your comments in italic font and then present our responses to the comments. 

Comment 1:Language improvement: The current draft is difficult to comprehend due to poor English. It is crucial to thoroughly revise and polish the language to ensure clarity and readability.

Response 1:The writer has revised the article for language issues.

Comment 2:Proper citation: It is essential to include citations for all the following papers:

- Bao, Z., & Huang, D. (2020). Gender differences in reaction to enforcement mechanisms: A large-scale natural field experiment.

- Bao, Z., & Huang, D. (2021). Shadow banking in a crisis: Evidence from FinTech during COVID-19. Journal of Financial and Quantitative Analysis, 56(7), 2320–2355.

- Bao, Z., & Huang, D. (2022a). Can Artificial Intelligence Improve Gender Equality? Evidence from a Natural Experiment.

- Bao, Z., & Huang, D. (2022b). Reform scientific elections to improve gender equality. Nature Human Behaviour, 6(4), 478–479.

- Bao, Z., & Huang, D. (2023). Gender-specific favoritism in science. Journal of Economic Behavior & Organization.

- Chen, M., Li, N., Zheng, L., Huang, D., & Wu, B. (2022). Dynamic correlation of market connectivity, risk spillover and abnormal volatility in stock price. Physica A: Statistical Mechanics and Its Applications, 587, 126506.

- Chen, M., Wang, Y., Wu, B., & Huang, D. (2021). Dynamic analyses of contagion risk and module evolution on the SSE a-shares market based on minimum information entropy. Entropy, 23(4), 434.

- Huang, D. (2020). How effective is social distancing. Covid Economics, Vetted and Real-Time Papers (59), 118–148.

- Li, N., Chen, M., Gao, H., Huang, D., & Yang, X. (2023). Impact of lockdown and government subsidies on rural households at early COVID-19 pandemic in China. China Agricultural Economic Review, 15(1), 109–133.

- Li, N., Chen, M., & Huang, D. (2022). How Do Logistics Disruptions Affect Rural Households? Evidence from COVID-19 in China. Sustainability, 15(1), 465.

- Wu, B., Huang, D., & Chen, M. (2023). Estimating contagion mechanism in global equity market with time-zone effect. Financial Management, 52, 543–572.

- Yu, D., & Huang, D. (2023a). Cross-sectional uncertainty and expected stock returns. Journal of Empirical Finance, 72, 321–340.

- Yu, D., Huang, D., & Chen, L. (2023). Stock return predictability and cyclical movements in valuation ratios. Journal of Empirical Finance, 72, 36–53.

- Yu, D., Huang, D., Chen, L., & Li, L. (2023). Forecasting dividend growth: The role of adjusted earnings yield. Economic Modelling, 120, 106188.

Response 2:The authors have been cited according to the needs of the article.

---

## [Decision Letter · Decision Letter 2]

3 Apr 2024

PONE-D-23-28533R2Capital market liberalization and corporate environmental performance: Evidence from the Shanghai–Hong Kong Stock ConnectPLOS ONE

Dear Dr. Zhong,

Thank you for submitting your manuscript to PLOS ONE. After careful consideration, we feel that it has merit but does not fully meet PLOS ONE’s publication criteria as it currently stands. Therefore, we invite you to submit a revised version of the manuscript that addresses the points raised during the review process.

We look forward to receiving your revised manuscript.

Kind regards,

Rana Muhammad Ammar Zahid, PhD

Academic Editor

PLOS ONE

Additional Editor Comments:

One or more of the reviewers has recommended that you cite specific previously published works. Members of the editorial team have determined that the works referenced are not directly related to the submitted manuscript. As such, please note that it is not necessary or expected to cite the works requested by the reviewer. 

The reviewer raised many issue in the current draft and i agree with him. Please incorporate the suggested changes.

Reviewers' comments:

Reviewer's Responses to Questions

**Comments to the Author**

1. If the authors have adequately addressed your comments raised in a previous round of review and you feel that this manuscript is now acceptable for publication, you may indicate that here to bypass the “Comments to the Author” section, enter your conflict of interest statement in the “Confidential to Editor” section, and submit your "Accept" recommendation.

Reviewer #1: All comments have been addressed

Reviewer #2: (No Response)

Reviewer #3: All comments have been addressed

2. Is the manuscript technically sound, and do the data support the conclusions?

Reviewer #1: Yes

Reviewer #2: Partly

Reviewer #3: Yes

3. Has the statistical analysis been performed appropriately and rigorously? 

Reviewer #1: Yes

Reviewer #2: No

Reviewer #3: Yes

4. Have the authors made all data underlying the findings in their manuscript fully available?

Reviewer #1: Yes

Reviewer #2: No

Reviewer #3: Yes

5. Is the manuscript presented in an intelligible fashion and written in standard English?

Reviewer #1: Yes

Reviewer #2: Yes

Reviewer #3: Yes

6. Review Comments to the Author

Reviewer #1: (No Response)

Reviewer #2: The implementation of the Shanghai–Hong Kong Stock Connect marks the maturity of China's capital market, and its effects have been the focus of academic attention. Based on this quasi-natural experiment, we selected 3248 samples of heavily-polluting enterprises listed in China from 2010 to 2020 to examine the impact of capital market liberalization on corporate environmental performance. Results show that capital market liberalization significantly improved heavily-polluting enterprises’ environmental performance. Heterogeneity analysis results indicate that this positive effect varied across firms with different ownership and internal controls types. Finally, mechanism analysis results found that capital market liberalization promoted heavily-polluting firms’ environmental performance by increasing environmental assets, reducing stock price volatility, and improving information disclosure quality.

The work in this paper is interesting. However, we have several concerns that may be addressed before the paper can proceed.

1. The contribution of this study is not clear, and the introduction section is relatively brief. Sha et al. (2022), Zhang et al. (2023), Deng et al. (2023), Sha et al., 2022, and He et al. (2024) have thoroughly demonstrated the relationship between capital market opening and environmental performance, and have concluded that capital market opening improve environmental performance. Therefore, this greatly undermines the theoretical contribution of this study. This makes it difficult for the first innovation point of this study to continue, in other words, this work does not substantively contribute to the existing literature. Although this study attempts to differentiate from existing literature through mechanism research (environmental assets, stock price volatility, and information disclosure quality), these mechanisms can essentially be explained by the mitigation of information asymmetry (Zhang et al., 2022; Liu & Niu, 2023; Yi, 2023), investor environmental concern (Sha et al., 2022; Deng et al., 2023) and information disclosure (Qing, 2022) proposed in existing literature, weakening the contribution of this study. The authors need to elaborate on the differences with existing literature in detail and briefly outline the introduction section, which includes motivation, theoretical support, brief literature/literature gap, research objectives, research contributions, brief findings within a paragraph, and research structure.

2. The research motivation of this study is insufficient. Based on the realistic background, the authors need to identify the problems existing in environmental performance in China in the introduction section and explain why the Shanghai-Hong Kong Stock Connect can alleviate these problems to a certain extent, that is, to elucidate the causal logic between the Shanghai-Hong Kong Stock Connect and corporate environmental performance. Considering the literature background, the research topic, research methods, and research conclusions of this study do not fill the gaps in existing literature. Therefore, the authors need to delve deeper into the relationship between capital market opening and environmental performance and identify more specific issues or problems that have been overlooked by the academic community.

3. In terms of hypothesis development, this study lacks a comprehensive literature review and theoretical foundation and does not elaborate on the relationship between capital market opening and the mechanism variables. Therefore, we recommend that the authors cite relevant literature on the economic consequences of capital market opening (Chen et al., 2023; Wang et al., 2024) and literature on factors influencing green innovation (Takalo et al., 2021; Tseng et al., 2013; Xiang et al., 2022) to incorporate research conclusions into the logical framework of this study.

4. There are several issues in the empirical test of this study: (1) The time of enterprises' entry into the Shanghai-Hong Kong Stock Connect and the Shenzhen-Hong Kong Stock Connect differs. Therefore, the paper presents a staggered Difference-in-Differences (DID) research question, and this paper's approach of using 2014 as the base year is unsound. (2) The robustness tests of the paper are relatively weak, lacking thorough robustness tests. (3) Furthermore, the paper investigates the relationship between capital market liberalization and environmental performance by selecting heavily polluting enterprises among Chinese listed companies, which introduces sample selection bias. We suggest the author adopts the Heckman two-stage method to address endogeneity and employ alternative explanations to avoid the impact of concurrent policy shocks.

5. In the mechanism testing section, the authors need to present the measurement methods for each mechanism variable and propose relevant hypotheses for the mechanisms in the literature review and hypotheses.

6. This study lacks explanations for the regression coefficients and economic significance of the results. In the heterogeneity analysis, the authors need to present in the main text why the regression results show heterogeneity in different groups, that is, the inherent connection between the grouping variable and the baseline regression. Additionally, the study did not test the differences in coefficients between the two groups, which makes it difficult to rigorously prove the heterogeneity of the results between the two groups.

7. The conclusion section of this study is relatively brief and needs to provide policy recommendations based on the regression results.

Reference:

[1] Sha, Y., Zhang, P., Wang, Y., & Xu, Y. (2022). Capital market opening and green innovation——Evidence from Shanghai-Hong Kong stock connect and the Shenzhen-Hong Kong stock connect. Energy Economics, 111, 106048.

[2] Zhang, R., Fu, W., & Lu, T. (2023). Capital market opening and corporate environmental performance: Empirical evidence from China. Finance Research Letters, 53, 103587.

[3] Wang, F., Ma, J., & Liao, C. (2023). Capital market opening up and corporate green technology innovation: Evidence from China’s Stock Connect program. Applied Economics, 1-17.

[4] Zhang, P., Sha, Y., Wang, Y., & Wang, T. (2022). Capital market opening and stock price crash risk–Evidence from the Shanghai-Hong Kong stock connect and the Shenzhen-Hong Kong stock connect. Pacific-Basin Finance Journal, 76, 101864.

[5] Liu, M., & Niu, X. (2023). The impact of capital market opening on earnings management: Empirical evidence based on “Land− Port Connection”. Finance Research Letters, 103864.

[6] Yi, Y. (2023). Does the liberalization of stock market optimize firm's debt? Evidence from a policy experiment in China. Finance Research Letters, 58, 104395.

[7] Deng, P., Wen, J., He, W., Chen, Y. E., & Wang, Y. P. (2023). Capital market opening and ESG performance. Emerging Markets Finance and Trade, 59(13), 3866-3876.

[8] Wang, W., & Qu, Z. (2024). Capital market opening and commercial bank risk: Evidence from “Shanghai–Hong Kong Stock Connect”. Finance Research Letters, 59, 104827.

[9] Chen, G., & Wang, M. (2023). Stock market liberalization and earnings management: Evidence from the China–Hong Kong Stock Connects. Finance Research Letters, 58, 104417.

[10] Takalo, S. K., & Tooranloo, H. S. (2021). Green innovation: A systematic literature review. Journal of Cleaner Production, 279, 122474.

[11] Tseng, M. L., Wang, R., Chiu, A. S., Geng, Y., & Lin, Y. H. (2013). Improving performance of green innovation practices under uncertainty. Journal of cleaner production, 40, 71-82.

[12] Xiang, X., Liu, C., & Yang, M. (2022). Who is financing corporate green innovation?. International Review of Economics & Finance, 78, 321-337.

[13] He, J., Huang, R., Ding, J., Liu, Y., & Zhou, R. (2024). The Impact of Capital Market Opening on Enterprise Green Technology Innovation: Insights from the Shanghai–Hong Kong Stock Connect. Sustainability, 16(6), 2369.

[14] Qing, L. (2022). The impact of environmental information disclosure on Chinese firms' environmental and economic performance in the 21st century: a systematic review. IEEE Engineering Management Review, 50(4), 203-214.

Reviewer #3: The revision is largely satisfactory. This paper is suitable for publication in this journal. I have no further comments.

7. PLOS authors have the option to publish the peer review history of their article (what does this mean?). If published, this will include your full peer review and any attached files.

Reviewer #1: No

Reviewer #2: No

Reviewer #3: No

---

## [Author Response · Author response to Decision Letter 2]

10 Jul 2024

Comment 1:

The contribution of this study is not clear, and the introduction section is relatively brief. Sha et al. (2022), Zhang et al. (2023), Deng et al. (2023), Sha et al., 2022, and He et al. (2024) have thoroughly demonstrated the relationship between capital market opening and environmental performance, and have concluded that capital market opening improve environmental performance. Therefore, this greatly undermines the theoretical contribution of this study. This makes it difficult for the first innovation point of this study to continue, in other words, this work does not substantively contribute to the existing literature. Although this study attempts to differentiate from existing literature through mechanism research (environmental assets, stock price volatility, and information disclosure quality), these mechanisms can essentially be explained by the mitigation of information asymmetry (Zhang et al., 2022; Liu & Niu, 2023; Yi, 2023), investor environmental concern (Sha et al., 2022; Deng et al., 2023) and information disclosure (Qing, 2022) proposed in existing literature, weakening the contribution of this study. The authors need to elaborate on the differences with existing literature in detail and briefly outline the introduction section, which includes motivation, theoretical support, brief literature/literature gap, research objectives, research contributions, brief findings within a paragraph, and research structure.

Response 1:

 Thanks for your great suggestion on improving the accessibility of our manuscript. We have added the relevant literature.The details are as follows:

Previous studies have also examined the relationship between market liberalization and environmental performance[5-6]. However, their selection of samples and the quantification of environmental performance lacked specificity. This research is based on heavily polluting enterprises as the research samples, so the sample selection is more targeted. Besides，we use emission charges and expenditures on environmental taxes to quantify corporate environmental performance, so the research results will be closely related to its business-level environmental performance. 

Comment 2:

The research motivation of this study is insufficient. Based on the realistic background, the authors need to identify the problems existing in environmental performance in China in the introduction section and explain why the Shanghai-Hong Kong Stock Connect can alleviate these problems to a certain extent, that is, to elucidate the causal logic between the Shanghai-Hong Kong Stock Connect and corporate environmental performance. Considering the literature background, the research topic, research methods, and research conclusions of this study do not fill the gaps in existing literature. Therefore, the authors need to delve deeper into the relationship between capital market opening and environmental performance and identify more specific issues or problems that have been overlooked by the academic community.

Response 2:

Investors use corporate environmental performance as a crucial indicator to understand companies’ sustainable development[3],evaluate investment value[4], and reflect the corporate value in the stock price. Before liberalization, Chinese government has developed many legal systems for environmental protection, but environmental problems remain serious. The reasons for the above problems are mainly manifested in the following aspects.On the one hand, some enterprises lack sufficient motivation to reduce the emission of pollutants facing pressure from the market, technology and capital. On the other hand some enterprises do not have enough knowledge about the environmental pollution problems, which leads to the ineffective solution. The liberalization of capital market provides an opportunity to solve the problem of environmental pollution. Some studies have demonstrated that capital market liberalization can improve corporate environmental performance[5-6], but there is a lack of specificity in sample selection and the conclusions may lack applicability. This study examines the relationship between capital market liberalization and corporate environmental performance using heavily polluting firms as research samples. Results show that capital market liberalization significantly enhances corporate environmental performance. The mechanism of action test indicates that capital market liberalization enhances corporate environmental performance by enhancing investment in environmental assets, reducing stock volatility, and improving the quality of information disclosure. The heterogeneity test finds that capital market liberalization has significant effect on corporate environmental performance in Non-SOEs and firms with low internal control.

Comment 3:

In terms of hypothesis development, this study lacks a comprehensive literature review and theoretical foundation and does not elaborate on the relationship between capital market opening and the mechanism variables. Therefore, we recommend that the authors cite relevant literature on the economic consequences of capital market opening (Chen et al., 2023; Wang et al., 2024) and literature on factors influencing green innovation (Takalo et al., 2021; Tseng et al., 2013; Xiang et al., 2022) to incorporate research conclusions into the logical framework of this study.

Response 3:

 In Literature Review,we have added an analysis of the relevant literature.The details are as follows:

2.1.1 Capital market liberalization

Existing literature on the economic consequences of capital market liberalization focuses mainly on the operation of capital markets,the behavior of enterprises, and the development of the real economy.First,capital market liberalization has introduced mature value investors to developing countries[6], which can guide investors to make value investments[7], improve the information content of stock prices[8], reduce the heterogeneous volatility of stock markets[9], and thus improve stock market stability and market effectiveness[10].Second, market transactions conducted by foreign investors can affect corporate decision-making behavior, such as improving the efficiency of corporate investment[6], reducing investment in inefficient labor[9],reducing commercial bank risk[11], reducing firms’earnings management[12],and promoting corporate social responsibility[13].Finally, foreign investors can optimize corporate governance mechanisms to curb aggressive financial asset allocation by enterprises and enhance the quality of financial services to the real economy[14].

2.1.2 Corporate environmental performance

Firstly, the factors influencing the corporate environmental performance.In relevant studies on corporate environmental performance,some studies are conducted at the macro level,Such as political connections[15],environmental protection tax[16],Climate risk[17],Low-carbon city pilot policy[18],and environmental regulation[19].Other studies focus on micro-level factors,Such as powerful CEOs[20], Multiple large shareholders[21];digital investment[22],corporate social responsibility[23],and genetic diversity on corporate boards[24].Second, the importance of corporate environmental performance.Some scholars argue that green innovation (GI) can be used as an alternative to corporate environmental performance.Takalo et al.(2021) suggest that A large number of organizations and communities have been directed towards green innovation as a strategy to achieve environmental protection and economic growth[25].Third, evaluating enterprises environmental performance .Tseng et al. (2013) assessed the green innovation practices of printed circuit board manufacturing companies in Taiwan in terms of four dimensions: management, process, product, and technological innovation,and also found that management innovation is a key driver of green innovation[26].

Comment 4:

 There are several issues in the empirical test of this study: (1) The time of enterprises' entry into the Shanghai-Hong Kong Stock Connect and the Shenzhen-Hong Kong Stock Connect differs. Therefore, the paper presents a staggered Difference-in-Differences (DID) research question, and this paper's approach of using 2014 as the base year is unsound. (2) The robustness tests of the paper are relatively weak, lacking thorough robustness tests. (3) Furthermore, the paper investigates the relationship between capital market liberalization and environmental performance by selecting heavily polluting enterprises among Chinese listed companies, which introduces sample selection bias. We suggest the author adopts the Heckman two-stage method to address endogeneity and employ alternative explanations to avoid the impact of concurrent policy shocks.

Response 4:

First , this study selects the companies of Shanghai-Hong Kong Stock Connect as the research sample, and does not select the companies of Shenzhen-Hong Kong Stock Connect as the research sample. The opening of Shanghai-Hong Kong Stock Connect was in 2014.

Second，We added the Heckman test.

4.3.6 Heckman test

 We use the Heckman two-stage model to control for the problem of self-selection of samples. Referring to Zhang (2023),In addition to the original control variables, we add variables that may affect the selection of firms into the SH-HKSC , including firm value (TobinQ), Listage(Age), stock turnover(Turnover), and dividend payout ratio (Dividen). The first stage probit regression results are shown in column 1 of Table 8. The results of Heckman's second stage regression are in Column 2 and 3 of Table 8, and the inverse Mills coefficient (imr) is significant at the 1 per cent level.The regression coefficients for List*Post remain significant when we control for relevant control variables, time effects, and firm effects, consistent with the baseline regression results.

Table 8. Heckman test

Variable SH-HKSC(1) EID(2) HEID(3)

List*Post 0.060***(11.05) -0.059***(-6.94)

Size 0.025*(1.20) 0.023**(3.06) 0.019*(2.64)

Sg 0.126**(4.28) 0.102*(2.65) 0.096*(4.36)

Intan 0.001(0.10) 0.001(0.00) 0.002(0.00)

LEV -0.032*(1.34) -0.114**(6.95) -0.157**(11.03)

DL 0.241***(11.52) 0.138***(16.30) 0.206***(15.30)

FS -0.057**(-5.03) -0.048*(-2.92) -0.062*(-3.61)

INS 0.359***(22.03) 0.267***(19.63) 0.168***(16.30)

TobinQ 0.143***(9.06) 

Age 0.010*(3.03) 

Turnover 0.031(1.37) 

Dividen 0.548***(33.64) 

imr 1.305***(36.02) 0.962***(30.87)

Year/Firm Fixed effect NO YES YES

R2 0.103 0.268 0.251

N 3248 3248 3248

Comment 5:

In the mechanism testing section, the authors need to present the measurement methods for each mechanism variable and propose relevant hypotheses for the mechanisms in the literature review and hypotheses.

Response 5:

First, we quantified each mediating variable.second，we propose relevant hypotheses for the mechanisms in the hypotheses.

Variable Name Variable Symbol Variable Definition

Corporate environmental performance EID the ratio of the logarithm of emissions charges to the logarithm of operating revenues

 HEID the ratio of the logarithm of corporate environmental taxes to the logarithm of operating revenues

Underlying Stocks of Shanghai-Hong Kong Stock Connect List List is a company dummy variable,which is 1 if the firm belong to the SH-HKSC,and 0 otherwise

Shanghai-Hong Kong Stock Connect Launch Time Post Post is a time dummy variable, which is 1 when year is after 2014，and 0 otherwise.

Environmental assets EA the ratio of environmental assets to total assets

Stock price volatility VOL the standard deviation of daily stock returns over the quarter

Information disclosure quality DQ DQ comes from CSMAR.The indicator is categorized into four levels and assigned values from 1 ( bad ) to 4 ( good ).

First,capital market liberalization affects corporate environmental performance through investing environmental assets.Investments in environmental assets are characterized by large investments, long lead times and low short-term benefits. As the social effect brought about by environmental asset investment is greater than the economic benefit, this also leads to the fact that in enterprises with high financing constraints, environmental asset investment will reduce the funds for other project investments, causing enterprise costs to rise and profits to decrease.Opening a capital market effectively integrates China's originally closed market with the global market,enhancing the efficiency of China's capital market[24,25].Good environmental performance can attract new investors ,and obtain sufficient funds to alleviate financing constraints.Feng et al. (2022) find that capital market liberalization reduces firms' financing constraints, thereby enhancing corporate green innovation.[26].Heavy polluting enterprises have enough money to invest in environmental assets to achieve lower pollutant emissions and environmental taxes, which in turn improves corporate environmental performance. 

Based on the above analysis, we propose the following hypothesis:

Hypothesis 1: Capital market liberalization enhances environmental performance by increasing investment in environmental assets.

Second, capital market liberalization leads to higher synchronization of China's capital market with the overseas capital market, which to a certain extent bears the impact of large-scale capital flows and thus affects capital market stability [27]. In particular, after international risk events, which include the U.S. subprime mortgage crisis and the 2015 stock market crash in China ,substantial capital outflows from the capital market cause dramatic fluctuations in the market, increasing companies short-term financial risks.Environmental performance, as an important strategic resource, can alleviate the external market’s impact on firms to some extent[28]. In other words,enterprises with good environmental performance can relieve stock price fluctuations to facilitate better access to funds for business growth through the capital markets[29,30,31].Thus, firms should have the motivation and capability to enhance environmental performance.

Hypothesis 2: Capital market liberalization enhances environmental performance by reducing stock price volatility

Finally, Capital market liberalization may facilitate the enhancement of corporate environmental performance by improving the information environment.Kim and Zhang(2014) find that when firms take on more social responsibility, they can curb the spread of negative news by increasing the transparency of financial reporting[32].Nie et al.(2023) find that Capital market liberalization enhances corporate ESG disclosure and satisfies the needs of investors[33]. Thus,the pressure to disclose environmental information motivates heavily polluting firms to enhance their environmental protection efforts, thereby improving corporate environmental performance.

Based on the above analysis, we propose the following hypothesis:

Hypothesis 3: Capital market liberalization improves corporate environmental performance by improving the quality of disclosure.

Comment 6:

This study lacks explanations for the regression coefficients and economic significance of the results. In the heterogeneity analysis, the authors need to present in the main text why the regression results show heterogeneity in different groups, that is, the inherent connection between the grouping variable and the baseline regression. Additionally, the study did not test the differences in coefficients between the two groups, which makes it difficult to rigorously prove the heterogeneity of the results between the two groups.

Response 6:

4.4. Heterogeneity analysis 

 Table 8 shows the regression results of the firm heterogeneity.We categorize firms into state-owned and non-state-owned firms(SOEs and Non-SOEs), and firms with high internal controls and low internal controls. The List*Post coefficients are significantly higher for Non-SOEs and firms with low internal control than for SOEs and firms with high internal control, respectively.Firstly, compared to SOEs, non-SOEs have more incentives to illustrate the effectiveness of their environmental protection through good environmental performance in order to gain government policy support and investors' attention.Secondly, firms with low internal con

---

## [Editor Report · Decision Letter 3]

25 Jul 2024

PONE-D-23-28533R3Capital market liberalization and corporate environmental performance: Evidence from the Shanghai–Hong Kong Stock ConnectPLOS ONE

Dear Dr. Zhong,

Thank you for submitting your manuscript to PLOS ONE. After careful consideration, we feel that it has merit but does not fully meet PLOS ONE’s publication criteria as it currently stands. Therefore, we invite you to submit a revised version of the manuscript that addresses the points raised during the review process.

We look forward to receiving your revised manuscript.

Kind regards,

Rana Muhammad Ammar Zahid, PhD

Academic Editor

PLOS ONE

Journal Requirements:

Additional Editor Comments:

Dear Dr. Zhong,

After careful consideration and review, I request minor amendments to this manuscript before accepting it for publication.

Please answer the reviewers comments and highlight the changes made in yellow or different font color to ease the review process and improve the manuscript, as it has area to improve in all section, especially introduction, and hypothesis development. Moreover, there is an recent literature with respect to the environment studies variables as discussed in following studies. These studies will help you to improve your arguments and methodology.

https://doi.org/10.1016/j.eap.2022.06.009

https://doi.org/https://doi.org/10.1111/beer.12607

https://doi.org/https://doi.org/10.1002/mde.4161

Make sure to proofread the manuscript before it is resubmitted to the journal. Please go through the journal’s guidelines thoroughly and revise the paper accordingly. Thank you for submitting your paper to the PlosOne.

---

## [Author Response · Author response to Decision Letter 3]

26 Sep 2024

Re: Response for manuscript”Capital market liberalization and corporate environmental performance - Evidence from the Shanghai-Hong Kong Stock Connect”

Dear Reviewers,

Thank you very much for your time involved in reviewing the manuscript and your very encouraging comments on the merits.

We also appreciate your clear and detailed feedback and hope that the explanation has fully addressed all of your concerns. In the remainder of this letter, we discuss each of your comments individually along with our corresponding responses. 

Comment 1:

Please answer the reviewers comments and highlight the changes made in yellow or different font color to ease the review process and improve the manuscript, as it has area to improve in all section, especially introduction, and hypothesis development. Moreover, there is an recent literature with respect to the environment studies variables as discussed in following studies. These studies will help you to improve your arguments and methodology.

Response 1:

 Thanks for your great suggestion on improving the accessibility of our manuscript. We have added the relevant literature.The details are as follows:

Investors use corporate environmental performance as a crucial indicator to understand companies’ sustainable development[3],evaluate investment value[4], and reflect the corporate value in the stock price. Before liberalization, Chinese government has developed many legal systems for environmental protection, but environmental problems remain serious. The reasons for the above problems are mainly manifested in the following aspects.On the one hand, some enterprises lack sufficient motivation to reduce the emission of pollutants facing pressure from the market, technology and capital. On the other hand some enterprises do not have enough knowledge about the environmental pollution problems, which leads to the ineffective solution. The liberalization of capital market provides an opportunity to solve the problem of environmental pollution.Some studies have demonstrated that capital market liberalization can promote corporate environmental performance[5-6], and there is also to be a study that shows that capital market liberalization capital market liberalization promotes green innovation in China and has a stronger impact on green invention patents than on green utility model patents[7],but there is a lack of specificity in sample selection and the conclusions may lack applicability. This study examines the relationship between capital market liberalization and corporate environmental performance using heavily polluting firms as research samples. Results show that capital market liberalization significantly enhances corporate environmental performance. The mechanism of action test indicates that capital market liberalization enhances corporate environmental performance by enhancing investment in environmental assets, reducing stock volatility, and improving the quality of information disclosure. The heterogeneity test finds that capital market liberalization has significant effect on corporate environmental performance in Non-SOEs and firms with low internal control.

2.1.2 Corporate environmental performance

Firstly, the factors influencing the corporate environmental performance.In relevant studies on corporate environmental performance,some studies are conducted at the macro level,Such as political connections[16],environmental protection tax[17],Climate risk[18],Low-carbon city pilot policy[19],and environmental regulation[20].Other studies focus on micro-level factors,Such as powerful CEOs[21], Multiple large shareholders[22];digital investment[23],corporate social responsibility[24], managers capabilities[25],board gender diversity[26],and genetic diversity on corporate boards[27].Zahid et al.(2024) suggested that managers with strong management skills exhibit a positive correlation with their company's ESG performance and its subcomponents[25].Zahid et al.(2023) found that more women who served on corporate boards enhanced the company's environmental performance and disclosures while limiting greenwashing behavior. Second, the importance of corporate environmental performance.Some scholars argue that green innovation (GI) can be used as an alternative to corporate environmental performance.Takalo et al.(2021) suggested that A large number of organizations and communities have been directed towards green innovation as a strategy to achieve environmental protection and economic growth[28].Third, evaluating enterprises environmental performance .Tseng et al. (2013) assessed the green innovation practices of printed circuit board manufacturing companies in Taiwan in terms of four dimensions: management, process, product, and technological innovation,and also found that management innovation is a key driver of green innovation[29]. 

We would like to take this opportunity to thank you for all your time involved and this great opportunity for us to improve the manuscript. We hope you will find this revised version satisfactory. 

Sincerely,

Zhong

---

## [Editor Report · Decision Letter 4]

8 Oct 2024

Capital market liberalization and corporate environmental performance: Evidence from the Shanghai–Hong Kong Stock Connect

PONE-D-23-28533R4

Dear Dr. Zhong,

We’re pleased to inform you that your manuscript has been judged scientifically suitable for publication and will be formally accepted for publication once it meets all outstanding technical requirements.

Kind regards,

Ömer Tuğsal Doruk

Academic Editor

PLOS ONE

Additional Editor Comments (optional):

Dear Authors,

As a result of the revisions, the work is suitable for publishing.

All the best,
---

## [Editor Report · Acceptance letter]

30 Oct 2024

PONE-D-23-28533R4 

PLOS ONE

Dear Dr. Zhong, 

I'm pleased to inform you that your manuscript has been deemed suitable for publication in PLOS ONE. Congratulations! Your manuscript is now being handed over to our production team.

Kind regards, 

on behalf of

Dr. Ömer Tuğsal Doruk 

Academic Editor

PLOS ONE